Accepted at the ICLR 2024 Workshop on AI4Differential Equations In Science

# Comparing and Contrasting Deep Learning Weather Prediction Backbones on Navier-Stokes Dynamics

**Matthias Karlbauer**[1,2*]  **Danielle C. Maddix**[1]  **Abdul Fatir Ansari**[1]  **Boran Han**[1]

**Gaurav Gupta**[1]  **Yuyang Wang**[1]  **Andrew Stuart**[3]  **Michael W. Mahoney**[4]

[1] Amazon Web Services (AWS)
[2] Neuro-Cognitive Modeling Group, University of Tübingen, Germany
[3] Amazon Search
[4] Amazon Supply Chain Optimization Technologies

## Abstract

There has been remarkable progress in the development of Deep Learning Weather Prediction (DLWP) models, so much so that they are poised to become competitive with traditional numerical weather prediction (NWP) models. Indeed, a wide number of DLWP architectures—based on various backbones, including U-Net, Transformer, Graph Neural Network (GNN), or Fourier Neural Operator (FNO)—have demonstrated their potential at forecasting atmospheric states. However, due to differences in training protocols, data choices (resolution, selected prognostic variables, or additional forcing inputs), and forecast horizons, it still remains unclear which of these methods and architectures is most suitable for weather forecasting. Here, we back up and provide a detailed empirical analysis, under controlled conditions, comparing and contrasting the most prominent backbones used in DLWP models. This is done by predicting two-dimensional incompressible Navier-Stokes dynamics with different numbers of parameters and Reynolds number values. In terms of accuracy, memory consumption, and runtime, our results illustrate various tradeoffs, and they show favorable performance of FNO, in comparison with Transformer, U-Net, and GNN backbones.

## 1 Introduction

Deep Learning Weather Prediction (DLWP) models have recently evolved to form a promising and competitive alternative to numerical weather prediction (NWP) models (Bauer et al., 2015). Early attempts (Scher & Messori, 2018; Weyn et al., 2019) designed U-Net models (Ronneberger et al., 2015) on a cylinder mesh, learning to predict air pressure and temperature dynamics on a coarse global resolution of $5.625°$. More recently, Pathak et al. (2022) designed FourCastNet on the basis of the Fourier Neural Operator (FNO) (Li et al., 2020b), moving to the native resolution of $0.25°$ of the ERA5 reanalysis dataset (Hersbach et al., 2020), which amounts to $721 \times 1440$ data points covering the globe. The same dataset finds application in the Vision Transformer (ViT) (Dosovitskiy et al., 2020) based Pangu-Weather model (Bi et al., 2023) and the state-of-the-art message-passing Graph Neural Network (GNN) (Battaglia et al., 2018; Fortunato et al., 2022) based GraphCast model (Lam et al., 2022).

In a comparison of state-of-the-art DLWP models, Rasp et al. (2023) find that GraphCast generates the most accurate weather forecasts on lead-times up to ten days. GraphCast was trained on 221 variables from ERA5 (substantially more than Pangu-Weather and FourCastNet, which were trained on 67 and 24 prognostic variables, respectively), but the root of GraphCast's improved performance

---

*Work done during an internship at AWS.

remains entangled in details of the architecture type, choice of prognostic variables, and training protocol. Here, we seek to elucidate the effect of the model architecture, i.e., GNN, Transformer, U-Net, or FNO, and design an empirical evaluation that thoroughly compares DLWP backbone architectures under controlled conditions while freezing data choice and training protocol. To this end, we train and evaluate various architectures on two-dimensional Navier-Stokes simulations, while controlling the number of parameters to generate cost-performance tradeoff curves. We motivate our data choice of using the Navier-Stokes equations since they find application in NWP where they are used to describe parts of the atmosphere's dynamics.

With this analysis, we seek to motivate a model that has the greatest potential in addressing downsides of current DLWP models, e.g., stability of long roll-outs for climate lengthscales, uncertainty quantification, and physically meaningful predictions. Our aim is to help the community find and agree on a suitable backbone for DLWP models.

## 2 RELATED WORK AND METHODS

We compare six model classes that form the basis for state-of-the-art DLWP models. We provide a brief overview of these methods in the following. (See Appendix A.1 for details and Table 2 in this appendix for how we modify them to vary the number of parameters.) As a naïve baseline and upper bound for our error comparison, we implement `Persistence`, which predicts the last observed state. Starting with early deep learning (DL) methods, we include convolutional long short-term memory (`ConvLSTM`) (Shi et al., 2015), which combines spatial and temporal information processing by replacing the scalar computations of LSTM gates (Hochreiter & Schmidhuber, 1997) with convolution operations. `ConvLSTM` is one of the first DL models for precipitation nowcasting and other spatiotemporal forecasting tasks, and it finds applications in Google's MetNet1 and MetNet2 (Sønderby et al., 2020; Espeholt et al., 2022). Among early DL methods, we also benchmark `U-Net`, which is one of the most prominent and versatile DL architectures, originally designed for biomedical image segmentation (Ronneberger et al., 2015) and forms the backbone of many DLWP models (Weyn et al., 2019; 2020; 2021; Karlbauer et al., 2023; Lopez-Gomez et al., 2023).

We also include two more recent architecture backbones based on Transformer and GNN, which power current state-of-the-art DLWP models. The Transformer architecture (Vaswani et al., 2017) has found success with image processing (Dosovitskiy et al., 2020) and has been applied to weather forecasting, by viewing the atmospheric state as a sequence of three-dimensional images (Gao et al., 2022). Pangu-Weather (Bi et al., 2023, by Huawei) and FuXi (Chen et al., 2023), use the `SwinTransformer` backbone (Liu et al., 2021), whereas Microsoft refines the Transformer into ClimaX, a model for weather and climate related downstream tasks (Nguyen et al., 2023). Multi-Scale MeshGraphNet (`MS MeshGraphNet`) (Fortunato et al., 2022) extends MeshGraphNet (Pfaff et al., 2020)—a message-passing GNN processing unstructured meshes—to operate on multiple grids with different resolutions. `MS MeshGraphNet` forms the basis of GraphCast (Lam et al., 2022) using a hierarchy of icosahedral meshes on the sphere.

Lastly, we benchmark architectures based on `FNO` (Li et al., 2020b). `FNO` is a type of operator learning method (Li et al., 2020a; Lu et al., 2021; Gupta et al., 2021) that learns a function-to-function mapping by combining pointwise operations in physical space and in the wavenumber/frequency domain. In contrast to the aforementioned architectures, it is a discretization invariant operator method. While `FNO` can be applied to higher resolutions than it was trained on, it may not be able to predict processes that unfold on smaller scales than observed during training (Krishnapriyan et al., 2023). These uncaptured small-scale processes can be important in turbulence modeling. We implement both a two- and a three-dimensional variant of `FNO`, as specified in Appendix A.1. We also experiment with `TFNO`, using a Tucker-based tensor decomposition (Tucker, 1966; Kolda & Bader, 2009), which is more parameter efficient. `FNO` serves as the basis for LBL's and NVIDIA's `FourCastNet` series (Pathak et al., 2022; Bonev et al., 2023; Kurth et al., 2023), which we also include in our comparison.[1]

---

[1]Due to the rectangular nature of our data, we consider the original FourCastNet implementation based on Guibas et al. (2021) instead of the newer Spherical Fourier Neural Operator (SFNO) (Bonev et al., 2023), which works with spherical data and is promising for weather prediction on the sphere.

Table 1: RMSE scores for experiment 1, reported for each model under different number of parameters. Errors reported in italic correspond to models that had to be retrained with gradient clipping (by norm) due to stability issues. With `OOM` and `sat`, we denote models that ran out of GPU memory and saturated, respectively. Saturated means that we did not further increase the parameters because the performance already saturated over smaller parameter ranges. Best results shown in bold.

| Model | #params | | | | | | | | |
|---|---|---|---|---|---|---|---|---|---|
| | 5 k | 50 k | 500 k | 1 M | 2 M | 4 M | 8 M | 16 M | 32 M |
| Persistence | .5993 | .5993 | .5993 | .5993 | .5993 | .5993 | .5993 | .5993 | .5993 |
| ConvLSTM | .1278 | .0319 | .0102 | *.0090* | *.2329* | *.4443* | OOM | —- | —- |
| U-Net | .5993 | .0269 | .0157 | .0145 | .0131 | .0126 | .0126 | sat | —- |
| FNO3D L1-8 | .3650 | .2159 | .1125 | .1035 | .1050 | .0383 | .0144 | .0095 | —- |
| TFNO3D L1-16 | —- | —- | —- | .0873 | .0889 | .0221 | .0083 | .0066 | .0069 |
| TFNO3D L4 | —- | .0998 | .0173 | .0127 | .0107 | .0091 | .0083 | sat | —- |
| TFNO2D L4 | **.0632** | **.0139** | **.0055** | **.0046** | **.0043** | .0054 | **.0041** | **.0046** | sat |
| SwinTransformer | .1637 | .0603 | .0107 | .0084 | .0070 | OOM | —- | —- | —- |
| FourCastNet | .1558 | .0404 | .0201 | .0154 | *.0164* | *.0153* | *.0149* | sat | —- |
| MS MeshGraphNet | *.2559* | *.0976* | *.5209* | OOM | —- | —- | —- | —- | —- |

## 3  EXPERIMENTS AND RESULTS

We conduct three series of experiments to explore the ability of the architectures (see Section 2) to predict the two-dimensional incompressible Navier-Stokes dynamics in a periodic domain. Our data is discretized on a two-dimensional $64 \times 64$ grid, and we design the experiments to test two levels of difficulties by generating less and more turbulent data, with Reynolds Numbers $Re = 1 \times 10^3$ (experiment 1) and $Re = 1 \times 10^4$ (experiments 2 and 3), respectively. For experiments 1 and 2, we generate $1\,\text{k}$ samples, while experiment 3 repeats experiment 2 with an increased number, $10\,\text{k}$, of samples. Our experiments are designed to test: (1) easier vs. harder problems, with the modification in $Re$, and (2) the effect of the data size. For comparability, the initial condition and forcing of the data generation process are chosen to be identical with those in Li et al. (2020b); Gupta et al. (2021). (See Appendix A.2.) Also, following Li et al. (2020b), the models receive a context history of $h = 10$ input frames, on basis of which they autoregressively generate the remaining 40 (experiment 1) or 20 (experiments 2 and 3) frames.[2] Concretely, we apply a rolling window when generating autoregressive forecasts, by feeding the most recent $h$ frames as input and predicting the next single frame, i.e., $\hat{y}_{t+1} = \varphi_\theta(x_{t-h,\ldots,t})$, where $\hat{y}_{t+1}$ denotes the prediction of the next frame generated by model $\varphi$ with trainable parameters $\theta$, and $x_{t-h,\ldots,t}$ denotes the most recent $h$ frames provided as input concatenated along the channel dimension. The three-dimensional (T)FNO models make an exception to the autoregressive rolling window approach, by receiving the first $h$ frames $x_{0:h}$ as input to directly generate a prediction $\hat{y}_{h+1:T}$ of the entire remaining sequence in a single step. See Appendix A.3 for our training protocol featuring hyperparameters, learning rate scheduling, and weight updates.

**Experiment 1: Reynolds Number $Re = 1 \times 10^3$, $1\,\text{k}$ samples.**  In this experiment, we generate less turbulent dynamics with Reynolds Number $Re = 1 \times 10^3$ and a sequence length of $T = 50$. The quantitative root mean squared error (RMSE) metric, reported in Table 1 and Figure 1 (left) shows that `TFNO2D` performs best, followed by `TFNO3D`, `SwinTransformer`, `FNO3D`, `ConvLSTM`, `U-Net`, `FourCastNet`, and `MS MeshGraphNet`. (See the qualitative results in Figure 2 in Appendix B.1 with the same findings.) All models outperform the naïve `Persistence` baseline, which always predicts the last observed state, i.e., $\hat{y}_t = x_h$. This principally indicates a successful training of all models. We observe substantial differences in the error saturation when increasing the number of parameters, which supports the ordering of architectures seen in Figure 2. Concretely, with an error of $1 \times 10^{-2}$, `MS MeshGraphNet` does not reach the accuracy level of the other models. Beyond $500\,\text{k}$ parameters, the model hits the memory constraint and also does not con-

---

[2]Larger Reynolds Numbers lead to more turbulent dynamics that are harder to predict. Thus, Li et al. (2020b) selects $T = 50$ and $T = 30$ for $Re = 1e3$ and $Re = 1e4$, respectively. We follow this convention.

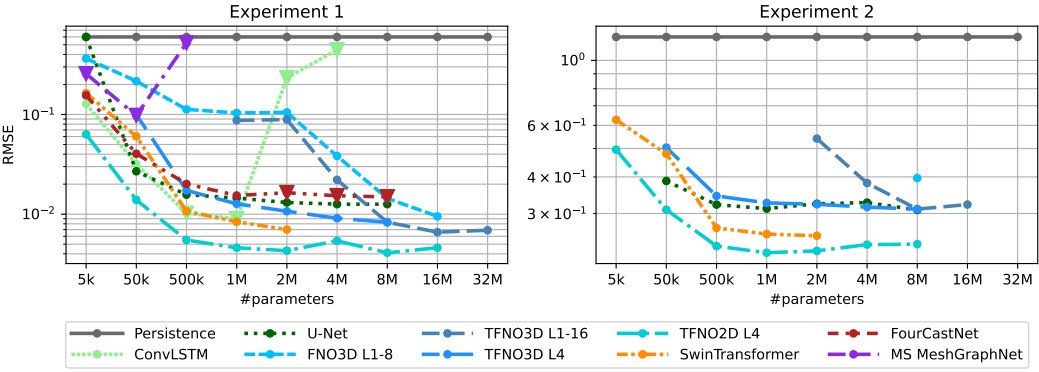

Figure 1: RMSE vs. number of parameters for models trained on Reynolds Numbers $Re = 1 \times 10^3$ (experiment 1, left) and $Re = 1 \times 10^4$ (experiment 2, right) with 1k samples. Note the different y-axis scales. Triangle markers indicate models with instability issues during training, requiring the application of gradient clipping. In the limit of growing parameters, each model converges to an individual error score (left), which seems consistent across data complexities (cf. left and right).

verge.[3] When investigating the source of this unstable training behavior, we identify remarkable effects of the graph design by comparing periodic 4-stencil, 8-stencil, and Delaunay triangulation graphs, where the latter supports a stable convergence most (see Figure 3 in Appendix B.1 for details). Throughout our experiments, we use the 4-stencil graph. `ConvLSTM` is competitive within the low-parameter-regime, saturating around an RMSE of $9 \times 10^{-3}$, and becomes unstable with large channel sizes (which we could not compensate even with gradient clipping), runs out of memory beyond $4\,\mathrm{M}$ parameters, and suffers from exponential runtime complexity (see Figure 4 (right) in Appendix B.1). Similarly, `SwinTransformer` generates comparably accurate predictions, reaching an error of $7 \times 10^{-3}$, before quickly running out of memory when going beyond $2\,\mathrm{M}$ parameters. `U-Net` and `FourCastNet` express a similar behavior, saturating at the $1\,\mathrm{M}$ parameter configuration and reaching error levels of $1.2 \times 10^{-2}$ and $1.5 \times 10^{-2}$, respectively. In `FNO3D` and `TFNO3D`, we observe a two-staged saturation, where the models first converge to a poor error regime of $1 \times 10^{-1}$, albeit approaching a remarkably smaller RMSE of $9 \times 10^{-3}$ and $6 \times 10^{-3}$, respectively, when increasing the number of layers from 1 at `#params` $\leq 2\,\mathrm{M}$ to 2, 4, 8, and 16 to obtain the respective larger parameter counts.[4] Instead, when fixing the numbers of layers at $l = 4$ and varying the number of channels in `TFNO3D L4`, we observe better performance compared to the single-layer `TFNO3D L1-16` in the low-parameter regime (until $2\,\mathrm{M}$ parameters), albeit not competitive with other models. To additionally explore the effect of the number of layers vs. channels in `TFNO3D`, we vary the number of parameters either by increasing the layers over $l \in [1, 2, 4, 8, 16]$, while fixing the number of channels at $c = 32$ in `TFNO3D L1-16`, or by increasing the number of channels over $c \in [2, 8, 11, 16, 22, 32]$ while fixing the number of layers at $l = 4$ in `TFNO3D 4L`. Consistent with Li et al. (2020b), we observe the performance saturating at four layers. Finally, the autoregressive `TFNO2D` performs remarkably well across all parameter ranges—converging at an unparalleled RMSE score of $4 \times 10^{-3}$—while, at the same time, constituting a reasonable trade-off between memory consumption and runtime complexity (see Figure 4 in Appendix B.1). From this we conclude that, at least for periodic fluid flow simulation, `FNO2D` marks a promising choice, suggesting its application to real-world weather forecasting scenarios.

**Experiment 2: Reynolds Number $Re = 1 \times 10^4$, 1k samples.** In this experiment, to stress test the consistency of the model order found in experiment 1, we generate more turbulent data by increasing the Reynolds Number $Re$ by an order of magnitude and reducing the simulation time and sequence length to $T = 30$ timesteps. To reduce our search over various architectures, we

---

[3]Experiments are performed on an AWS `g5.12xlarge` instance featuring four NVIDIA A10G GPUs with 23 GB RAM each. We used single GPU training throughout all our experiments.

[4]We observe a similar behavior (not shown) when experimenting with the number of blocks vs. layers in `SwinTransformer`, suggesting to prioritise more layers per block over more blocks with less layers.

focus here on stable models only by removing those that depend on gradient clipping. We make the same observations as in experiment 1, confirming `TFNO2D` as the most accurate model, followed by `SwinTransformer`, `TFNO3D`, and `U-Net` on this harder task. See Figure 1 (right) for quantitative results and Figure 6 in Appendix B.2 for qualitative results.

**Experiment 3: Reynolds Number $Re = 1 \times 10^4$, 10 k samples.** Since in experiment 2, the three-dimensional TFNO models with `#params` $\geq 8\,M$ start to show a slight tendency to overfit (not shown), we repeat the experiment, increasing the number of training samples by an order of magnitude to $10\,k$. Figure 5, Figure 7 and Table 4 in Appendix B.2 show that the same findings hold on experiment 3, which has a larger number of $10\,k$ samples.

## 4 DISCUSSION

In this work, we obtain insights into which DLWP models are more suitable for weather forecasting by devising controlled experiments, which are useful more generally, to compare these models. In particular, we fix the input data and training protocol and vary the architecture and number of parameters. In this limited setup on the synthetic two-dimensional periodic incompressible Navier-Stokes dataset, we find that `TFNO2D` performs the best at predicting the dynamics, followed by `TFNO3D`, `SwinTransformer`, `FNO3D`, `ConvLSTM`, `U-Net`, `FourCastNet`, and `MS MeshGraphNet`. Importantly, throughout our experiments, we see that all benchmarked models eventually saturate at an individual error regime, indicating that scaling laws could not be determined for the models and task at hand. We thus identify a value for further research in developing an architecture that scales with increased parameters without converging at a specific error level. The spherical Fourier Neural Operator SFNO (Bonev et al., 2023) marks a reasonable starting point for further exploration, ranging from increasing the number of prognostic variables, to providing more context to the models by means of forcing fields, to incorporating physical knowledge (Li et al., 2022; Hansen et al., 2023) and moving beyond deterministic forecasts (Cachay et al., 2023; Gao et al., 2023; Price et al., 2023).

A limitation of our work is the sole consideration of synthetic data. We seek to focus our future endeavors on repeating the analysis presented here on WeatherBench (Rasp et al., 2020) to verify our claims and suggestions on real-world weather data that is represented on the sphere. In addition, although we enable circular padding in the compared architectures, the periodic nature of the data we use may favor the inductive bias of FNO. The periodicity exhibited in our data matches that of weather dynamics on global scale. Throughout our experiments, we perform only single model runs. When experimenting with different seeds, we do not encounter severe variations, which is also supported by the consistent RMSE vs. parameter plots. We note the poor results for `MS MeshGraphNet`, which we attribute to the underlying graph and to our training protocol. Seeing that GraphCast has been trained very carefully on 221 prognostic variables, its lead over other DLWP may be dominated by the number of variables and by the training protocol, and less to the actual architecture. This hypothesis, however, needs further investigations in future work such as on WeatherBench.

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

## A  MODEL, DATA, AND TRAINING SPECIFICATIONS

In this section, we discuss the model configurations and how we vary the number of parameters in our experiments. In addition, we detail the dataset generation and training protocols.

### A.1  MODEL CONFIGURATIONS

We compare six model classes that form the basis for state-of-the-art DLWP models. We provide details about each model and how we modify them in order to vary the number of parameters below. Table 2 provides an overview and summary of the parameters and model configurations.

**ConvLSTM**  We first implement an encoder—to increase the model's receptive field—consisting of three convolutions with kernel size $k = 3$, stride $s = 1$, padding $p = 1$, set padding_mode = circular to match the periodic nature of our data, and implement $\tanh$ activation functions. We add four ConvLSTM cells, also with circular padding and varying channel depth (see Table 2 for details), followed by a linear output layer. Being the only recurrent model, we

Table 2: Model configurations partitioned by model and number of parameters (which amount to the trainable weights). For configurations that are not specified here, the default settings from the respective model config files are applied, e.g., ConvLSTM employs the default from configs/model/convlstm.yaml, while overriding hidden_sizes by the content of the "Dim." column of this table. Details are also reported in the respective model paragraphs of Appendix A.1.

| Model | #params | Enc. | Dim. | Dec. |
|---|---|---|---|---|
| ConvLSTM | 5 k | $3 \times$ Conv2D with tanh() | $4 \times 4$ | $1 \times$ Conv2D |
| | 50 k | | $4 \times 13$ | |
| | 500 k | | $4 \times 40$ | |
| | 1 M | | $4 \times 57$ | |
| | 2 M | | $4 \times 81$ | |
| | 4 M | | $4 \times 114$ | |
| | 8 M | — | — | — |

| Model | Dim. (hidden sizes) |
|---|---|
| U-Net | $[1, 2, 4, 8, 8]$ |
| | $[3, 6, 12, 24, 48]$ |
| | $[8, 16, 32, 64, 128]$ |
| | $[12, 24, 48, 96, 192]$ |
| | $[16, 32, 64, 128, 256]$ |
| | $[23, 46, 92, 184, 368]$ |
| | $[33, 66, 132, 264, 528]$ |

| Model | #params | #modes | Dim. | #layers |
|---|---|---|---|---|
| (T)FNO3D L1-16 | 5 k | $3 \times 3$ | 11 | 1 |
| | 50 k | $3 \times 3$ | 32 | 1 |
| | 500 k | $3 \times 7$ | 32 | 1 |
| | 1 M | $3 \times 10$ | 32 | 1 |
| | 2 M | $3 \times 12$ | 32 | 1 |
| | 4 M | $3 \times 12$ | 32 | 2 |
| | 8 M | $3 \times 12$ | 32 | 4 |
| | 16 M | $3 \times 12$ | 32 | 8 |
| | 32 M | $3 \times 12$ | 32 | 16 |

| Model | #modes | Dim. | #layers |
|---|---|---|---|
| TFNO2D L4 | $2 \times 12$ | 2 | 4 |
| | $2 \times 12$ | 8 | 4 |
| | $2 \times 12$ | 27 | 4 |
| | $2 \times 12$ | 38 | 4 |
| | $2 \times 12$ | 54 | 4 |
| | $2 \times 12$ | 77 | 4 |
| | $2 \times 12$ | 108 | 4 |
| | $2 \times 12$ | 154 | 4 |
| | — | — | — |

| Model | #params | #modes | Dim. | #layers |
|---|---|---|---|---|
| TFNO3D L4 | 5 k | — | — | — |
| | 50 k | $3 \times 12$ | 8 | 4 |
| | 500 k | $3 \times 12$ | 11 | 4 |
| | 1 M | $3 \times 12$ | 16 | 4 |
| | 2 M | $3 \times 12$ | 22 | 4 |
| | 4 M | $3 \times 12$ | 32 | 4 |
| | 8 M | $3 \times 12$ | 45 | 4 |

| Model | Dim. | #layers |
|---|---|---|
| FourCastNet | 12 | 1 |
| | 64 | 1 |
| | 112 | 4 |
| | 160 | 4 |
| | 232 | 4 |
| | 326 | 4 |
| | 468 | 4 |

| Model | #params | #heads | Dim. | #blocks | #lrs/blck |
|---|---|---|---|---|---|
| Swin-Transformer | 5 k | 1 | 8 | 1 | 1 |
| | 50 k | 2 | 8 | 2 | 2 |
| | 500 k | 4 | 40 | 2 | 4 |
| | 1 M | 4 | 60 | 2 | 4 |
| | 2 M | 4 | 88 | 2 | 4 |

| Model | $D_{processor}$ | $D_{other}$ |
|---|---|---|
| MS Mesh-GraphNet | 8 | 8 |
| | 34 | 32 |
| | 116 | 32 |
| | — | — |
| | — | — |

perform ten steps of teacher forcing before switching to closed loop to autoregressively unroll a prediction into the future.

**U-Net**   We implement a five-layer encoder-decoder architecture with avgpool and transposed convolution operations for down and up-sampling, respectively. On each layer, we employ two consecutive convolutions with ReLU activations (Fukushima, 1975) and apply the same parameters described above in the encoder for `ConvLSTM`. See Table 2 for the numbers of channels hyperparameter setting.

**SwinTransformer**   Enabling circular padding and setting patch size $p = 2$, we benchmark the shifted window transformer (Liu et al., 2021) by varying the number of channels, heads, layers, and blocks, as detailed in Table 2, while keeping remaining parameters at their defaults.

**MS MeshGraphNet**   We formulate a periodically connected graph to apply Multi-Scale Mesh-GraphNet (`MS MeshGraphNet`) with two stages, featuring 1-hop and 2-hop neighborhoods, and follow Fortunato et al. (2022) by encoding the distance and angle to neighbors in the edges. We employ four processor and two node/edge encoding and decoding layers and set $\text{hidden\_dim} = 32$ for processor, node encoder, and edge encoder, unless overridden (see Table 2).

**FNO**   We compare three variants of `FNO`: Two three-dimensional formulations, which process the temporal and both spatial dimensions simultaneously to generate a three-dimensional output of shape $[T, H, W]$ in one call, and a two-dimensional version, which only operates on the spatial dimensions of the input and autoregressively unrolls a prediction into the future. While fixing the lifting and projection channels at 256, we vary the number of Fourier modes, channel depth, and number of layers according to Table 2.

**FourCastNet**   We choose a patch size of $p = 4$, fix $\text{num\_blocks} = 4$, enable periodic padding in both spatial dimensions, and keep the remaining parameters at their default values while varying the number of layers and channels as specified in Table 2.

## A.2   DATA GENERATION

We provide additional information about the data generation process in Table 3, which we keep as close as possible to that reported in Li et al. (2020b) and Gupta et al. (2021).

## A.3   TRAINING PROTOCOL

In the experiments, we use the Adam optimizer with learning rate $\eta = 1 \times 10^{-3}$ (except for `MS MeshGraphNet`, which only converged with a smaller learning rate of $\eta = 1 \times 10^{-4}$) and cosine learning rate scheduling to train all models with a batch size of $B = 4$, effectively realizing $125\,\text{k}$

Table 3: Settings for training, validation, and test data generation in the experiments, where $f, T, \delta_t$, and $\nu$ denote the dynamic forcing, sequence length (corresponding to the simulation time, which, in our case, matches the number of frames, i.e., $\Delta t = 1$), time step size for the simulation, and viscosity (which is the inverse of the Reynolds Number, i.e., $Re = 1/\nu$), respectively. The parameters $\alpha$ and $\tau$ parameterize the Gaussian random field to sample an initial condition (IC) resembling the first timestep.

| | | | Simulation parameters | | IC | | #samples | | |
|---|---|---|---|---|---|---|---|---|---|
| Experiment | $f$ | $T$ | $\delta_t$ | $\nu$ | $\alpha$ | $\tau$ | Train | Val. | Test |
| 1 | * | 50 | $1 \times 10^{-2}$ | $1 \times 10^{-3}$ | 2.5 | 7 | 1000 | 50 | 200 |
| 2 | * | 30 | $1 \times 10^{-4}$ | $1 \times 10^{-4}$ | 2.5 | 7 | 1000 | 50 | 200 |
| 3 | * | 30 | $1 \times 10^{-4}$ | $1 \times 10^{-4}$ | 2.5 | 7 | 10000 | 50 | 200 |

$^*f = 0.1(\sin(2\pi(x + y)) + \cos(2\pi(x + y)))$, with $x, y \in [0, 1, \ldots, 63]$.

weight update steps, relating to 500 and 50 epochs, respectively, for $1\,\mathrm{k}$ and $10\,\mathrm{k}$ samples.[5] For the training objective and loss function, we choose the mean squared error (MSE) between the model outputs and respective ground truth frames, that is $\mathcal{L} = \mathrm{MSE}(\hat{y}_{h+1:T}, y_{h+1:T})$. Note that, to stabilize training, we have to employ gradient clipping (by norm) for selected models, indicated by italic numbers in tables and triangle markers in figures.

## B   ADDITIONAL RESULTS AND MATERIALS

In this section, we provide additional empirical results for the three experiments.

### B.1   RESULTS FROM EXPERIMENT 1: REYNOLDS NUMBER $Re = 1 \times 10^3$

Figure 2 illustrates the initial and end conditions along with the respective predictions of all models. Qualitatively, we find there exist parameter settings for all models to successfully unroll a plausible prediction of the Navier-Stokes dynamics over 40 frames into the future, as showcased by the last predicted frame, i.e., $\hat{y}_{t=T}$ (see the third and fifth row of Figure 2). When computing the difference between the prediction and ground truth, i.e., $d = \hat{y} - y$, we observe clear variations in the accuracy of the model outputs, denoted by the saturation of the difference plots in the second and fourth row of Figure 2. Interestingly, this difference plot also reveals artifacts in the outputs of selected models: `SwinTransformer` and `FourCastNet` generate undesired patterns that resemble their windowing and patching mechanisms, whereas the 2-hop neighborhood, which was chosen as the resolution of the coarser grid, is baked into the output of `MS MeshGraphNet`. According to the

Table 4: RMSE scores partitioned by experiments and reported for each model under different numbers of parameters. Errors reported in italic correspond to models that had to be retrained with gradient clipping (by norm) due to stability issues. With `OOM` and `sat`, we denote models that ran out of GPU memory and saturated, meaning that we did not train models with more parameters because the performance already saturated over smaller parameter ranges. Best results are reported in bold. More details about architecture specifications are reported in Appendix A.1 and Table 2.

| | Model | #params | | | | | | | | |
| | | 5 k | 50 k | 500 k | 1 M | 2 M | 4 M | 8 M | 16 M | 32 M |
|---|---|---|---|---|---|---|---|---|---|---|
| **Experiment 1** | Persistence | .5993 | .5993 | .5993 | .5993 | .5993 | .5993 | .5993 | .5993 | .5993 |
| | ConvLSTM | .1278 | .0319 | .0102 | *.0090* | *.2329* | *.4443* | OOM | —– | —– |
| | U-Net | .5993 | .0269 | .0157 | .0145 | .0131 | .0126 | .0126 | sat | —– |
| | FNO3D L1-8 | .3650 | .2159 | .1125 | .1035 | .1050 | .0383 | .0144 | .0095 | —– |
| | TFNO3D L1-16 | —– | —– | —– | .0873 | .0889 | .0221 | .0083 | .0066 | .0069 |
| | TFNO3D L4 | —– | .0998 | .0173 | .0127 | .0107 | .0091 | .0083 | sat | —– |
| | TFNO2D L4 | **.0632** | **.0139** | **.0055** | **.0046** | **.0043** | **.0054** | **.0041** | **.0046** | sat |
| | SwinTransformer | .1637 | .0603 | .0107 | .0084 | .0070 | OOM | —– | —– | —– |
| | FourCastNet | .1558 | .0404 | .0201 | .0154 | *.0164* | *.0153* | *.0149* | sat | —– |
| | MS MeshGraphNet | *.2559* | *.0976* | *.5209* | OOM | —– | —– | —– | —– | —– |
| **Experiment 2** | Persistence | 1.202 | 1.202 | 1.202 | 1.202 | 1.202 | 1.202 | 1.202 | 1.202 | 1.202 |
| | U-Net | —– | .3874 | .3217 | .3117 | .3239 | .3085 | sat | —– | —– |
| | TFNO3D L1-8 | —– | —– | —– | —– | .5407 | .3811 | .3105 | .3219 | sat |
| | TFNO3D L4 | —– | .5038 | .3444 | .3261 | .3224 | .3155 | .3105 | sat | —– |
| | TFNO2D L4 | **.4955** | **.3091** | **.2322** | **.2322** | **.2236** | **.2349** | **.2358** | sat | —– |
| | SwinTransformer | .6266 | .4799 | .2678 | .2552 | .2518 | OOM | —– | —– | —– |
| **Experiment 3** | Persistence | 1.202 | 1.202 | 1.202 | 1.202 | 1.202 | 1.202 | 1.202 | 1.202 | 1.202 |
| | U-Net | —– | .3837 | .3681 | .2497 | .3162 | .2350 | .2383 | sat | —– |
| | TFNO3D L1-16 | —– | —– | —– | —– | .5146 | .2805 | .1814 | .1570 | .1709 |
| | TFNO3D L4 | —– | .4799 | .2754 | .2438 | .2197 | .2028 | .1814 | .1740 | sat |
| | TFNO2D L4 | **.4846** | **.2897** | **.1778** | **.1585** | **.1449** | **.1322** | **.1248** | **.1210** | sat |
| | SwinTransformer | .6187 | .4698 | .2374 | .2078 | .1910 | OOM | —– | —– | —– |

---

[5]With an exception for `MS MeshGraphNet`, which only supports a batch size of $B = 1$, resulting in $500\,\mathrm{k}$ weight update steps.

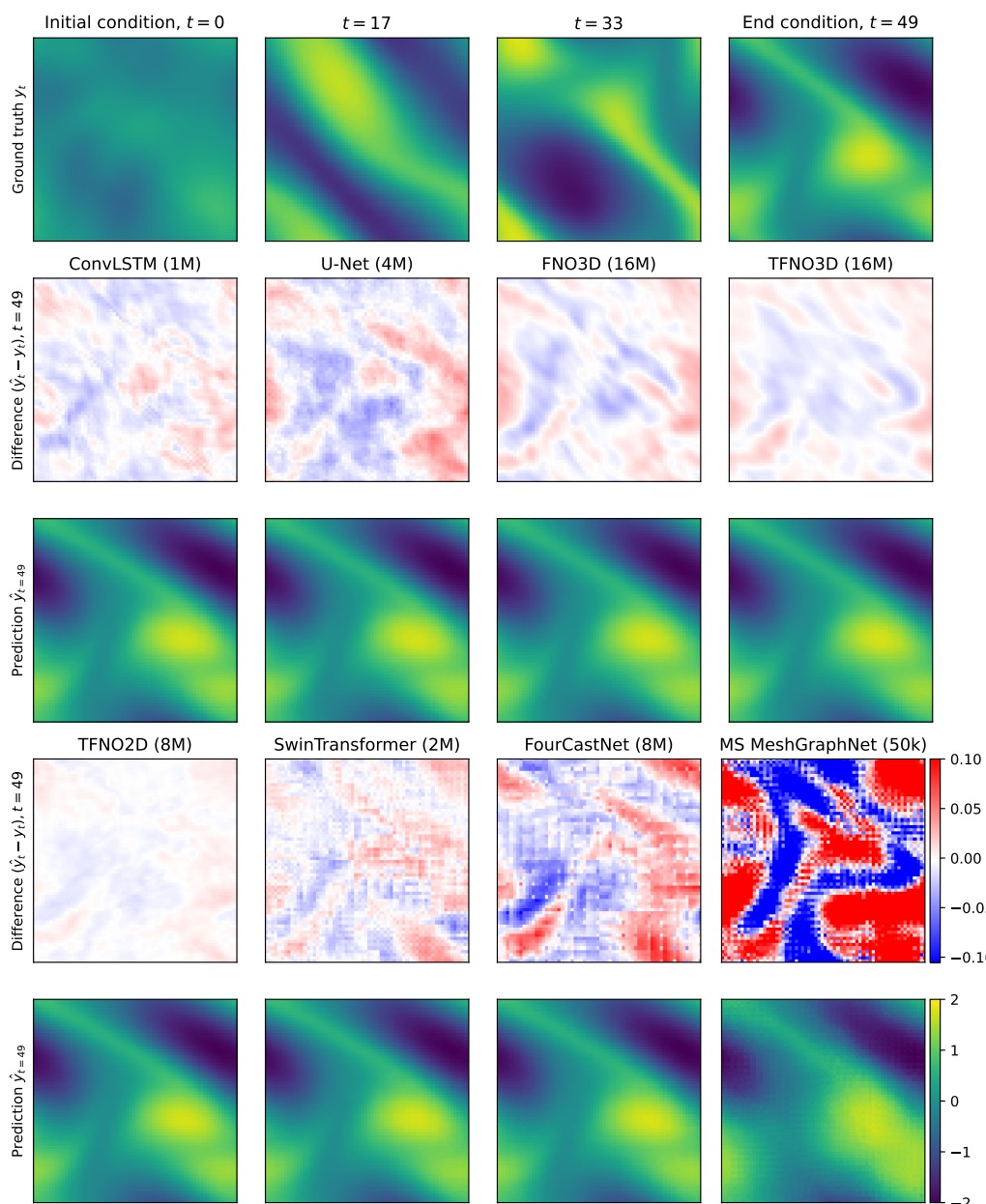

Figure 2: Qualitative results on the Navier-Stokes dataset with Reynolds Number $Re = 1 \times 10^3$ trained on $1\,\mathrm{k}$ samples (experiment 1). The first row shows the ground truth at four different points in time. The remaining rows show the difference between the predicted- and ground-truth at final time (row two and four), as well as the predicted final frame (row three and five). All models receive the first 10 frames of the sequence to predict the remaining 40 frames. The last frame of the predicted sequence from the best models are visualized and respective parameter counts are displayed in parenthesis.

lowest error scores reported in Table 4, we only visualize the best performing model among all parameter ranges in Figure 2 and observe the trend that TFNO2D performs best, followed by TFNO3D, SwinTransformer, FNO3D, ConvLSTM, U-Net, FourCastNet, and MS MeshGraphNet.

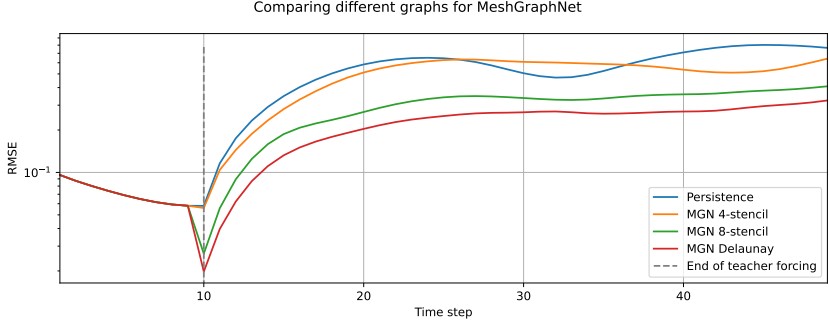

Figure 3: RMSE evolving over forecast time for three different underlying graphs (meshes) that are used in the single scale MeshGraphNet (MGN) (Pfaff et al., 2020).

Next, we study the effect of the underlying graph in GNNs. Observing the poor behavior of `MS MeshGraphNet` in Figure 2, we investigate the effect of three different periodic graph designs to represent the neighborhoods in the GNN. First, the 4-stencil graph connects each node's perpendicular four direct neighbors (i.e., north, east, south, and west) in a standard square Cartesian mesh. Second, the 8-stencil graph adds the direct diagonal neighbors to the 4-stencil graph. Third, the Delaunay graph connects all nodes in the graph by means of triangles, resulting in a hybrid of the 4-stencil and 8-stencil graph, where only some diagonal edges are added. To simplify the problem, we conduct this analysis on the single-scale MeshGraphNet (Pfaff et al., 2020) instead of using the hierarchical `MS MeshGraphNet` (Fortunato et al., 2022). While the graphs have the same number of nodes $|\mathcal{N}| = 4096$, their edge counts differ to $|\mathcal{E}_4| = 16384$, $|\mathcal{E}_8| = 32768$, and $|\mathcal{E}_D| = 24576$ for the 4-stencil, 8-stencil, and Delaunay graph, respectively. The results reported in this paper are based on the 4-stencil graph.

Interestingly, as indicated in Figure 3, the results favor the Delaunay graph over the 8- and 4-stencil graphs, respectively. Apparently, the increased connectedness is beneficial for the task. At the same time, though, the irregularity introduced by the Delaunay triangulation potentially forces the model to develop more informative codes for the edges to represent direction and distance of neighbors more meaningfully.

Lastly, Figure 4 compares the RMSE, memory consumption and computational cost in seconds per epoch as a function of the number of parameters. We see that `TFNO2D L4` performs the best in terms of the RMSE and also scales well with respect to memory and runtime.

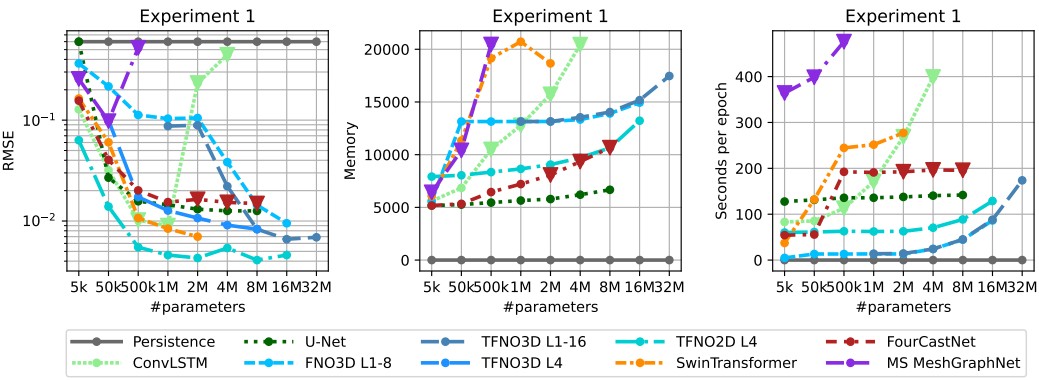

Figure 4: RMSE (left), memory consumption (center), and runtime complexity in seconds per epoch (right) over different parameter counts for models trained on Reynolds Number $Re = 1 \times 10^3$ with $1\,\mathrm{k}$ samples for experiment 1.

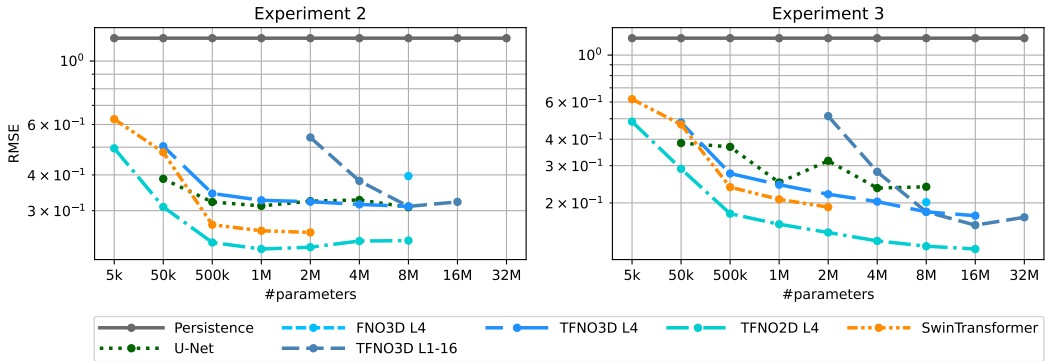

Figure 5: RMSE vs. parameters for models trained on Reynolds Number $Re = 1 \times 10^4$ with $1\,\mathrm{k}$ (experiment 2, left) and $10\,\mathrm{k}$ (experiment 3, right) samples. Note the different y-axis scales. Main observation: As expected, model performance correlates with the number of samples. The number of samples, though, does not affect the model ranking.

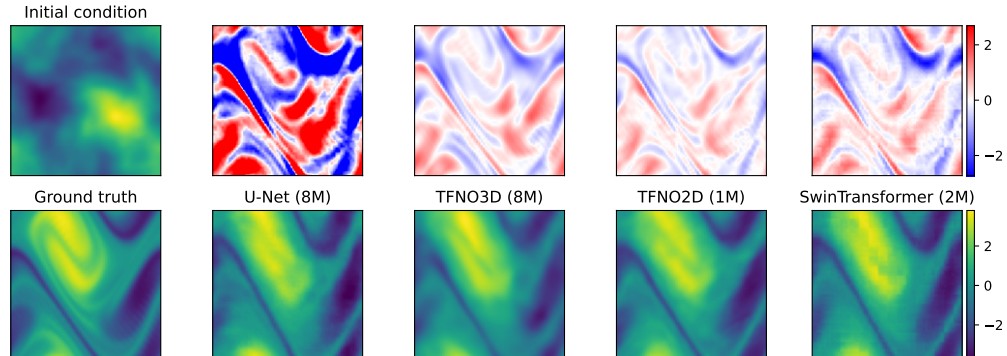

Figure 6: Qualitative results on Navier-Stokes data with Reynolds Number $1 \times 10^4$ trained on $1\,\mathrm{k}$ samples (experiment 2). The top left shows the initial condition. The remaining columns in the top row show the differences between the predicted and ground-truth at the final time for the various models. The bottom left shows the ground truth at the final time. The remaining columns in the bottom row show the final predictions from the various models to visually compare to the ground truth. All models face difficulties at resolving the yellow vortex, resulting in blurry predictions around the turbulent structure at this higher Reynolds Number. Among the parameter ranges, the best models are selected for visualizations (parameter count in brackets).

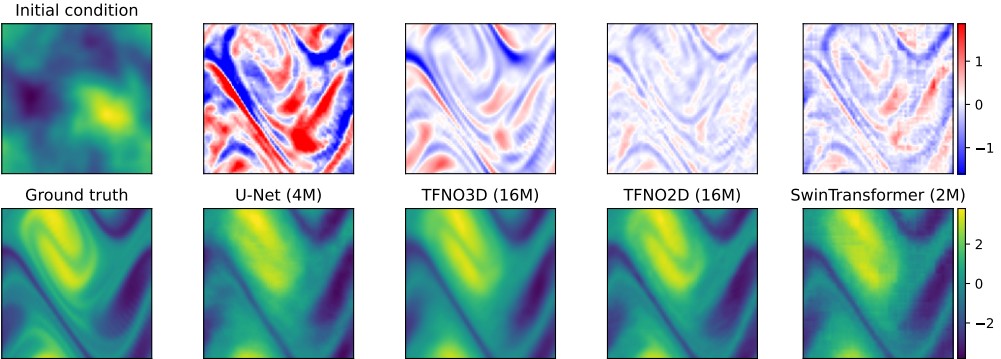

Figure 7: Qualitative results on Navier-Stokes data with Reynolds Number $1 \times 10^4$ trained on $10\,\mathrm{k}$ samples (experiment 3). In comparison to Figure 6, the yellow vortex is captured more accurately by `TFNO2D` as a consequence of the larger training set. See plot description in Figure 6 for details.

## B.2    RESULTS FROM EXPERIMENT 2 AND EXPERIMENT 3: $Re = 1 \times 10^4$

Table 4 shows the quantitative error scores of all the experiments (for an easier comparability). We see that the same trend occurs across all three experiments with `TFNO2D` performing the best. Figure 5 illustrates the similar trends of these RMSE results from experiments 2 and 3. Figure 6 and Figure 7 provide the qualitative visualizations for experiments 2 and 3, respectively. Figure 5 (right) and Figure 7 for experiment 3 show that, while all models consistently improve their scores due to the larger training set, the results from experiments 1-2 still hold. That is, when comparing the convergence levels in Figure 5 (right) and Table 4, we see that all models saturate at lower error regimes, while the ordering of the model performance from experiment 1 remains unchanged. Also qualitatively, the models benefit from the increase of training samples in experiment 3, since, visually, the yellow vortex at this higher Reynolds Number is resolved more accurately when the models are trained on more data, as illustrated in Figure 6 and Figure 7.

