# OpenReview forum: "Comparing and Contrasting Deep Learning Weather Prediction Backbones on Navier-Stokes Dynamics"
_ICLR.cc/2024/Workshop/AI4DiffEqtnsInSci — AI4DiffEqtnsInSci @ ICLR 2024 Poster_

### Official Review · Reviewer_K6Hb · 2024-02-22
**Benchmarking Deep Learning Weather Prediction Models**

**Rating:** 6
**Confidence:** 4

**Review:**

Summary:
The authors set out to benchmark various Deep Learning Weather Prediction (DLWP) models on a two-dimensional Navier Stokes problem, which they claim is a suitable problem for evaluating models of this kind.

Strengths and Weaknesses:

[+] The paper is well-motivated and clearly presented. It provides a clear description of the problem and the proposed study.

[+] The authors benchmark a wide variety of DLWP models, including not only different models but also different variations of models such as two- and three-dimensional TFNOs. The inclusion of a baseline (Persistence) and a multiple of parameters for each is very comprehensive.

[-] The claim that the chosen test problem is suitable for identifying a backbone for DLWP is let down by the author's own comments. Take, for example, footnote 1: "Due to the rectangular nature of our data, we consider the original FourCastNet implementation based on Guibas et al. (2021) instead of the newer Spherical Fourier Neural Operator (SFNO) ". For suitable benchmarking, it does not seem this is a sufficient problem; it would be better to go to 3D and on global scales (e.g., Figures 2, 6, & 7 in the appendix show small-scale features unlike the problems solved in something like the original FourCastNet paper) in addition to including a wider variety of problems. While this is an okay first step, drawing conclusions from such a small problem set does not seem prudent.

[+] The supplementary material is beneficial and includes many important and helpful details. However, the call for papers states: "Reviewers are not obliged to read supplementary materials when reviewing the paper." Some information in the supplementary material is necessary to corroborate the main text claims (see next comment).

[-] It would be nice to see training and inference times in addition to accuracy. The authors make the statement: "In terms of accuracy, memory consumption, and runtime, our results illustrate various tradeoffs, and they show favorable performance of FNO, in
comparison with Transformer, U-Net, and GNN backbones." in the abstract. However, I do not see any reporting of memory consumption or runtime in the main text, and inference time comparisons are also vital for real-time predictions. The authors do state when certain models run out of memory, but that is not comparative; it is only a binary statement.

Conclusion: In terms of models, the study is comprehensive; however, in terms of problems, it is not. Given this is a workshop paper, that is to be expected in only 4 pages, however, the conclusions derived from such a limited study have to be taken with a grain of salt. I think benchmarking lends itself better to a longer format venue, so comparisons and conclusions can be fair. Still, there are insights that can be made here, such as studying memory and saturation limitations on an individual model level; therefore, I am recommending a weak acceptance and would like to see this expanded upon in a full-length article.

---

### Official Review · Reviewer_YfR6 · 2024-02-26
**The Fourier Neural Operator shows favorable performance in predicting atmospheric states under controlled conditions compared to other Deep Learning Weather Prediction models.**

**Rating:** 9
**Confidence:** 4

**Review:**

The work discusses the recent advancements in Deep Learning Weather Prediction (DLWP) models, highlighting their potential to rival traditional numerical weather prediction (NWP) models. Various DLWP architectures, including U-Net, Transformer, Graph Neural Network (GNN), and Fourier Neural Operator (FNO), have shown promise in forecasting atmospheric states. However, due to differences in training protocols, data choices, and forecast horizons, it remains unclear which method and architecture are most suitable for weather forecasting. The study conducts a detailed empirical analysis comparing these architectures under controlled conditions, focusing on predicting two-dimensional incompressible Navier-Stokes dynamics. Results indicate favorable performance of the Fourier Neural Operator (FNO) backbone in terms of accuracy, memory consumption, and runtime when compared to Transformer, U-Net, and GNN backbones.

In general, the present study is of interest for the DL-based weather forecasting community, but also in particular for benchmarking for the machine learning community. The aim of the paper is clearly defined in the manuscript. The text is technically well-written and follows a clear structure and logic. The theoretical concepts are concisely presented in the main text, with further details described in the appendices. The figures and the experiments are in general sound, and the results support the conclusions. While these conclusions are based on synthetic data as a first step, and it is clear limitation of this work, it is fair to point to future work to confirm or challenge these results based on real-world weather data sets.

---

### Official Review · Reviewer_GapD · 2024-02-27
**Review of “Comparing And Contrasting Deep Learning Weather Prediction Backbones On Navier-Stokes Dynamics”**

**Rating:** 4
**Confidence:** 5

**Review:**

This work compares a range of state-of the art machine learning methods used for the emulation of weather prediction systems. This is a rapidly growing field with several different groups producing highly skilled models, all built using different core architectures. This paper’s novelty is directly comparing the skill of each architecture on a single system using similar training setups for each model to reduce the factors contributing to skill. The paper looks at three configurations of the system and training data to compare results. The topic of the paper is interesting as a comparison like this has yet to be performed on data from a climate system.

However, several flaws with the idealized testbed and training/testing methodology bring into question the usefulness of the paper’s results.

Major Concerns
1)	The two cases of homogenous isotropic forced 2D turbulence selected for the paper lack the multi-scale complexity present in true atmospheric flows. The flows selected are borderline turbulent, in addition to lacking other important characteristics such as rotation or mean flows.
a.	In experiment 1, with Reynold number of 1000, the flow looks to be close to laminar, with a single pair of vortices dominating the domain and no visible small scales. Given that the initial condition for the simulation appears to be from the spin up of a simulation, it is also possible the run has not had sufficient time to develop turbulence.
b.	In experiment 2, the flow does appear to be turbulent, however it still is dominated by a small number of large vortices.
2)	Choices the authors make while training the model may impact each architecture differently and differ to the implementations of the corresponding methods by the groups who have shown success with them.
a.	An example, providing the model with the previous 10 time-steps of history is not a common practice in weather predictions and may have both positive or negative effects on model prediction. These effects also may be a function of architecture.
3)	Ultimately the paper draws conclusions outside of the scope of the work done, investing more time into understanding the key features of climate/weather emulation and choosing a more suitable testbed (e.g. higher Reynolds numbers, including rotation, or even dynamics across multiple depths/layers as in quasi geostrophic models) would make this an important contribution to the field.
Minor Concerns
1)	The persistence RMSE is computed against a snapshot of undeveloped flow, this does not give a good baseline for the performance of the models as the system has not yet reached a steady state. In fully developed flow (which is the case for the atmosphere), the large vortices would persist longer and likely would give the system a longer decorrelation timescale and meaningfully change the values of RMSE.
a.	The use of persistence as an initial condition is not the most representative of problems in the atmosphere as learning how flow will develop is not the same as how it will evolve.
2)	Be careful with language in the paper, some statements are not correct.
a.	“the atmosphere is modeled by means of incompressible Navier-Stokes equations” – the atmosphere is not considered to be an incompressible flow, typically it is assumed to be an ideal gas in the dynamical core of climate models. Furthermore, the actual atmosphere reflects dynamics including moisture and radiation, which are not components of the Navier-Stokes equations.

---

### Meta-Review · Area_Chair_758A · 2024-03-01

**Recommendation:** Accept (Poster)

**Metareview:**

Thanks to reviewers for their careful and detailed comments. I also agree that this work could be considered as one preliminary step towards evaluating DLWP models. I agree though that using simplified benchmarks authors should not make strong conclusion without having enough data points and for that I encourage them to revisit and revise their conclusions appropriately based on the points addressed by reviewers. Also, reporting and or at least adding a short discussion on runtime and memory consumption of different methods is a great point by reviewer K6Hb, as it plays a role in the applicability of methods for real weather prediction problem. Authors are expected to address all the comments by reviewers in their final version and under that condition, I vote accept for the publication of this study as poster.

---

### Decision · Program_Chairs · 2024-03-03

Accept (Poster)